# Mesenchymal Stem Cells from COPD Patients Are Capable of Restoring Elastase-Induced Emphysema in a Murine Experimental Model

**DOI:** 10.3390/ijms24065813

**Published:** 2023-03-18

**Authors:** Carlos Río, Andreas K. Jahn, Aina Martin-Medina, Alba Marina Calvo Bota, Mª Teresa De Francisco Casado, Pere Joan Pont Antona, Orlando Gigirey Castro, Ángel Francisco Carvajal, Cristina Villena Portella, Cristina Gómez Bellvert, Amanda Iglesias, Javier Calvo Benito, Antoni Gayà Puig, Luis A. Ortiz, Ernest Sala-Llinàs

**Affiliations:** 1Inflammation, Repair and Cancer of Respiratory Diseases (i-Respire), Fundació Institut d’ Investigació Sanitària Illes Balears (IdISBa), 07120 Palma, Spain; 2Estabulary, Scientific-Technical Services, Universitat de les Illes Balears (UIB), 07122 Palma, Spain; 3Department of Thoracic Surgery, Hospital Universitari Son Espases, 07120 Palma, Spain; 4CIBERES Pulmonary Biobank Consortium, Hospital Universitari Son Espases, 07120 Palma, Spain; 5Department of pathology, Hospital Universitari Son Espases, 07120 Palma, Spain; 6Centro de Investigación Biomédica en Red de Enfermedades Respiratorias (CIBERES), Instituto de Salud Carlos III, 28029 Madrid, Spain; 7Banc de Teixits, Blood and Tissue Bank of the Balearic Islands (FBSTIB), 07120 Palma, Spain; 8Cell Therapy and Tissue Engineering Group (TERCIT), Institut d’ Investigació Sanitària Illes Balears (IdISBa), 07004 Palma, Spain; 9Department of Environmental and Occupational Health, University of Pittsburgh, Pittsburgh, PA 15260, USA; 10Department of Pulmonary Medicine, Hospital Universitari Son Espases, 07120 Palma, Spain

**Keywords:** COPD, MSC, elastase, pre-clinical

## Abstract

COPD is a chronic lung disease that affects millions of people, declining their lung function and impairing their life quality. Despite years of research and drug approvals, we are still not capable of halting progression or restoring normal lung function. Mesenchymal stem cells (MSC) are cells with extraordinary repair capacity, and MSC-based therapy brings future hope for COPD treatment, although the best source and route of administration are unclear. MSC from adipose tissue (AD-MSC) represents an option for autologous treatment; however, they could be less effective than donor MSC. We compared in vitro behavior of AD-MSC from COPD and non-COPD individuals by migration/proliferation assay, and tested their therapeutic potential in an elastase mouse model. In addition, we tested intravenous versus intratracheal routes, inoculating umbilical cord (UC) MSC and analyzed molecular changes by protein array. Although COPD AD-MSC have impaired migratory response to VEGF and cigarette smoke, they were as efficient as non-COPD in reducing elastase-induced lung emphysema. UC-MSC reduced lung emphysema regardless of the administration route and modified the inflammatory profile in elastase-treated mice. Our data demonstrate equal therapeutic potential of AD-MSC from COPD and non-COPD subjects in the pre-clinical model, thus supporting their autologous use in disease.

## 1. Introduction

Chronic obstructive pulmonary disease (COPD) is a lung disease with high morbidity and mortality that causes a global public health problem of huge magnitude [1]. It is characterized by chronic inflammation that leads to chronic bronchitis and loss of alveolar tissue, known as emphysema [2]. Current therapies are basically substantiated on inhaled bronchodilators and corticosteroids, or even triple therapy (long-acting muscarinic antagonist plus long-acting beta2-agonist plus inhaled corticosteroid), as recommended by the Global initiative for chronic Obstructive Lung Disease (GOLD) for patients that suffer from frequent exacerbations and dyspnoea [3]. Such treatments are effective in reducing dyspnoea and preventing exacerbations, which have contributed to reducing mortality; however, there is no treatment capable of halting the disease progression or restoring the lung [3]. Therefore, the development of therapies that are directed towards repair and regeneration are required. Increasing attention has been paid over recent years on the study of MSC as potential therapeutic agents for a wide compendium of respiratory diseases, including COPD [4,5]. MSC are multipotent cells which can be isolated from bone marrow (BM), adipose tissue (AD), and umbilical cord tissue (UC), among others. Besides their ability of differentiating into different cell lineages, MSC have raised growing interest because of their immunomodulatory and repair properties [6,7]. In recent years, a wide range of preclinical studies using rat/mouse models of COPD have shown that MSC from different origins are capable of reducing lung emphysema, attenuating pulmonary inflammation, and decreasing epithelial cell damage [4].

Bone marrow-derived MSCs (BM-MSCs) were the first MSCs to be discovered and used, and have long been considered the main source of MSCs for clinical applications, reporting that MSC administration is safe and has a systemic anti-inflammatory effect in COPD patients [4,8,9,10]. However, MSCs have subsequently been isolated from different sources, including umbilical cord (UC-MSC) and adipose tissue (AD-MSC) [11]. UC-MSC can be easily isolated in large numbers, have a stable surface marker expression in early passages (passages 4 ± 8), and the expression of immune response-related surface antigens, such as CD40, CD40 ligand, CD80, and CD86, is absent, facilitating the escape from the host immune attack [12,13]. On the other hand, AD-MSCs can be isolated more easily and in greater quantities than BM-MSCs and, unlike UC-MSCs, AD-MSC can be isolated from COPD patients themselves. In addition, some clinical studies have shown that administration of BM-MSC in patients with emphysema is safe but could not report a clear beneficial effect [10,13].

MSC from adipose origin (AD-MSC) might represent a convenient source for their clinical use; however, there is the question whether autologous MSC from diseased patients could be influenced systemically by the disease, thus being less efficient than MSC coming from healthy donors [14]. To date, there are no preclinical studies that would compare the therapeutic potential of AD-MSC from patients with COPD or pulmonary emphysema versus healthy donors to address this question. In this sense, it has recently been shown that BM-MSC from patients with idiopathic pulmonary fibrosis (IPF) shows less reparative and anti-inflammatory capacity than that obtained from patients without IPF [15], questioning the option of autologous transplant in that disease. However, if differences in the AD-MSC therapeutic potential also occur in pulmonary emphysema, this has not yet been studied. Another controversial aspect is the route of administration. Several studies observing human MSC distribution after intravenous (i.v.) infusion have demonstrate that they can rapidly rise the lower airways of the lungs, which might be an advantage to treat lung disease [16]. On the other hand, intratracheal (i.t.) administration might represent an advantage to better assess the upper bronchial areas of the lung, which are also affected in COPD.

In order to compare the functionality of COPD and non-COPD AD-MSC, we initially tested their behavior in vitro by cell differentiation, migration and attachment assays; and, next, in vivo, using an elastase generated emphysema nude mouse model to assess whether MSCs obtained from adipose tissue (AD-MSC) from healthy donors or COPD patients present different therapeutical effects. We also tested if a potentially superior therapeutical type of MSC, human umbilical cord-derived mesenchymal stem cells (UC-MSC), granted better outcomes in the same emphysematous mouse model. Next, we evaluated the molecular profile of the mouse lungs treated with elastase previous or after UC-MSC cellular therapy. We also appraise different routes of cell inoculation and weigh up the potential of detecting exogenous human mitochondria in this elastase mouse model.

## 2. Results

### 2.1. AD-MSC from COPD Patients Present Different Functional Responses Than Those from Donors in Migration and Cigarette Smoke Exposure

We first investigated whether human AD-MSC derived from COPD patients had comparable functional properties to those derived from non-COPD donors. MSC multipotency was tested by cellular differentiation assays and reported that AD-MSC from both COPD and non-COPD donors were equally able to differentiate into adipocytes and osteoblasts after 21 days of culture under stimulation with FABP4 (for adipocytes) or osteocalcin (for osteoblasts) (Figure 1a). The migration capacity of MSC is key for their homing at the injured sites [17]; therefore, we also monitored the migratory response of the AD-MSC towards a media containing nutrients (10% FCS) by real-time cell assay up to 12 h. Interestingly, we observed that COPD-derived AD-MSC had significantly increased migratory capacity throughout the whole assay when compared to non-COPD (Figure 1b, *p* = 0.0182, *p* = 0.0007, and *p* = 0.0002 at 3, 6 and 9 h respectively). We then decided to investigate the migratory capacity in response to VEGF-121 (Vascular Endothelial Growth Factor 121) and found that non-COPD significantly increased their migration towards VEGF-121 stimuli whereas no response was observed in COPD-derived AD-MSC (Figure 1c). Finally, in order to expose the AD-MSC to the context of cigarette smoking, which is a major cause of lung emphysema, we incubated non-COPD and COPD AD-MSC into a cigarette smoking chamber and monitored the cell attachment over time up to 3 h. We found that both non-COPD and COPD AD-MSC detached in a great extent in response to cigarette smoke; however, the cell detachment was significantly greater in the COPD cohort (Figure 1d, non-COPD vs non-COPD + VEG-121; *p* = 0.0366). Taken together, these results indicate that there exist functional differences between COPD and non-COPD AD-MSC in terms of migration and response to cigarette smoke.

### 2.2. AD-MSC from COPD Patients Reduced Elastase-Induced Emphysema at the Same Extent as Those from Non-COPD

While trying to generate an in vivo COPD/emphysema pre-clinal model, we started using the BALB/c strain of mice. Nonetheless, when analyzing the histology of this typical strain, we observed significant inflammation caused just by a regular inoculation of human MSCs unmistakably as a response to the cell xenotransplantation (Appendix A). Consequently, we started using the immunodeficient strain Crl:NU(Ico)-Foxn1^nu^ (Swiss nude), which does not reject human transplantation. To establish the best route of cellular administration, we applied AD-MSCs through 3 different routes and observed that intracheal (i.t.) or intravenous (i.v.) shots have a more notable distribution of the cells than intraperitoneal (i.p.), and that i.t. was at least as efficient as i.v. Nevertheless, i.t. inoculation was technically less challenging in our hands, and also less cellular signal could be observed in other organs than the lung, thus targeting more accurately the organ of interest (Appendix A). We also detected that in order to develop a clear emphysema in the lungs of these immunodeficient mice, we needed a total of 4 doses applied every other day, plus i.t. inoculations of higher concentrated, 0.028 IU/g, elastase.

We then compared the therapeutic potential of AD-MSC from COPD versus non-COPD sources in the elastase-treated mice. Given our abovementioned results, we stuck to the i.t. route for the following in vivo experiment according to the design in Figure 2a. Mice were distributed in four groups: (I) control, (II) elastase, (III) elastase + non-COPD AD-MSC, and (IV) elastase + COPD AD-MSC (Figure 2b). Both AD-MSC from non-COPD and COPD donors were able to reduce by more than a half the emphysematic area in the elastase-treated mice (53.3 ± 2.9 and 55.3 ± 9.5 percentage reduction for non-COPD and COPD AD-MSC treatments, respectively) (Figure 2c,d). According to this results, AD-MSC from COPD patients are equally efficient when it comes to reducing lung emphysematic damage.

### 2.3. Administration of Human UC-MSC Improved Elastase-Induced Emphysema

Human MSC from the Wharton’s jelly of umbilical cords (UC-MSC) are thought to be a more multipotent kind of stem cells with a prospective use in respiratory pathologies like COPD [18]. Therefore, we tested the therapeutic efficiency of these MSCs on lung emphysema. We advocated for a i.t. route as the most effective for retaining the MSC signal within the lung’ however, in the literature, i.v. has been by far the most used when it comes to UC-MSC applications. Therefore, we decided to include this route and compare both i.v. and i.t. applications for the in vivo experiments involving UC-MSC treatments. Immunodeficient mice were given elastase at 4 consecutive doses and then received UC-MSC in two separate administrations (i.v. + i.v. or i.t. + i.t.) at the timepoints and groups indicated (Figure 3a,b). Digital quantification of emphysematous areas of the lungs evidenced that application of human UC-MSC by both i.v. and i.t. routes significantly reduced the damage in elastase-treated mice when compared to those that received only elastase (Figure 3c,d). Notably, UC-MSC treatment reduced emphysematous getting even closer to the levels of control lungs than those observed in our previous results involving AD-MSC.

### 2.4. Human UC-MSC Leave Mitochondrial DNA and Modified the Immunomodulatory Cytokine and Chemokine Profile in the Elastase-Treated Mouse Lungs

We further investigated the mechanisms implicated in the therapeutic effect that we observed after UC-MSC treatment in our elastase. This was based on the current evidence that mitochondria play a critical role in MSC-driven repair [19]. We attempted to detect the presence of human MSCs and human (MSC-derived) mitochondria by analysing the human genomic or mitochondrial DNA, respectively, in lung homogenates at day 12 post-elastase (7 days after the last MSC administration). Human genomic DNA (RPP30) was nearly undetectable (Figure 4a); however, human mitochondrial DNA was found in similar numbers after administration of UC-MSC regardless of i.v. or i.t. routes (Figure 4b,c). This finding was confirmed by the observation of human immunostained mitochondria within the parenchyma in proximity to the lung airways of the MSC-treated mice (Appendix A). We also analyzed changes in the levels of cytokines involved in immunomodulation and repair by performing a proteomic array of the lung homogenates. No changes were observed in the immunomodulatory cytokines IL28 and IL33, which remained downregulated after elastase treatment regardless of MSC treatment; however, the levels of the anti-inflamatory cytokine IL11 were restored and further upregulated. In addition, the chemokyne CCL21 was found upregulated in the elastase-treated mice and further increased in the MSC-treated mice. In contrast, CCL22, although unchanged in the elastase group, appeared downregulated in response to MSC. Myeloperoxidase (MPO) and Fibroblast Growth Factor 1 (FGF1), which mediate the inflammatory response, were found upregulated in the elastase mice treated with MSC, whereas no changes were observed in the inflammatory mediator RAGE (Receptor for Advanced Glycan End products). Interestingly, there was a partial restorage in the levels of the adipokine ADIPSIN in the MSC-treated mice (Figure 4d). In summary, these results are indicating that UC-MSC treatment confers mitochondria and drive molecular changes in the inflammatory mediators in the lungs of elastase-treated mice.

## 3. Discussion

Due to their immunomodulatory properties and their role in mediating repair and regeneration, there are ongoing efforts to implement the use of MSC treatment for the therapy of chronic lung diseases such as COPD. However, the selection of the best MSC source and administration route remains a challenge, and also, it is not clear whether autologous treatment with MSC from the diseased patient would be as efficient as the treatment with MSC from non-COPD donors. In this work, we attempted to give answer to these questions by performing in vitro and in vivo studies to test: 1. the in vitro functionality and the in vivo therapeutic potential of AD-MSC derived from COPD compared to non-COPD subjects, and 2. the effectiveness of in vivo treatment of elastase-treated mouse with UC-MSC depending on the route of administration and the molecular mechanisms involved.

We performed in vitro experiments to compare the functional behaviour of AD-MSC from COPD and non-COPD patients and observed that, despite both having the same multipotency potential, COPD-derived AD-MSC had a different migratory profile. VEGF is known to promote MSC motility [20] and is reduced in sputum in COPD patients, which has been correlated with alveolar destruction [21]; moreover, loss of VEGF in emphysema lungs leads to endothelial cell apoptosis [22] and its secretion by distal lung fibroblasts mediates remodeling processes in these areas [23]. We found that COPD-derived AD-MSC can migrate towards nutrients but show decreased migration in response to VEGF stimulus when compared to non-COPD. This might indicate that whereas the injured lung is trying to attract MSC via upregulating VEGF, the homing response of the MSC is impaired in COPD patients. We also exposed the AD-MSC groups to cigarette smoke and, in accordance with Wahl et al. [24], we found inducement of cell death upon exposure; additionally, we report that cell death was significantly higher in AD-MSC from COPD patients when compared to non-COPD as observed by rapid cell detachment, indicating that they are more sensitive to tobacco effects.

We then tested COPD and non-COPD AD-MSC therapeutic potential in vivo by administrating intratracheally two doses of cell suspensions and found that both cohorts were capable of reducing emphysema in our elastase mouse model, as demonstrated by reduction in more than 50% of the emphysematous areas. Other preclinical studies from Antunes et al. [25] and Fujioka et al. [26] support the therapeutic applicability of AD-MSC in reducing elastase-induced emphysema; however, they only tested AD-MSC from healthy human subjects. Two phase I/II clinical studies have started to evaluate the effect of autologous AD-MSC as single cells or in the form of stem vascular fraction (SVF) applied right after isolation intravenously in patients with COPD (NCT04433104 and NCT02216630 respectively). These trials do not include AD-MSC from healthy donors; however, our results are important indicators that such autologous treatments could be as efficient as those with allogenic AD-MSC coming from healthy subjects.

So far, some clinical trials using MSC from bone marrow origin (BM-MSC) in COPD patients have reported promising results. Weiss et al. reported that infusion of BM-MSC can overall reduce levels of circulating protein C-reactive protein [9] and improve lung function in those with baseline high CRP levels [27]. Antunes et al. administrated BM-MSC in combination with endobronchial valve surgery, and, apart from reduced circulating CRP levels, they also reported significant improvement in quality-of-life indicators; however, lung function remained unchanged [28]. In line with them, Armitage et al. also found a reduction in inflammatory mediators but no improvement in lung function after BM-MSC infusions in COPD patients [29]. Despite the use of BM-MSC possibly having a beneficial effect, it still requires invasiveness to reach the cells; umbilical cord origin is another possibility, apart from adipose tissue, for obtaining MSC. In this regard, another ongoing phase I/II clinical study, with no results posted yet, aims to test the safety and efficacy of MSC of umbilical cord origin administrated intravenously in two doses 3 months apart in patients with COPD (NCT04433104). In addition to our experiments with AD-MSC, we performed another set of in vivo experiments to compare the therapeutic potential of MSC from umbilical cord (UC) origin. Despite the fact that in our biodistribution assays we observed that AD-MSC signal appears stronger when applied intratracheally, in the literature, the i.v. route is by far the most used to apply UC-MSC. Therefore, we administrated UC-MSC i.t. as well as i.v. in our elastase model and compared their therapeutic potential. We found that both routes of administration were able to restore the levels of lung emphysema similar to the levels in control mice; moreover, although not statistically significant, i.t. reduced emphysema a 30% more than i.v. administration. This might be due to the shorter distance that MSC need to travel to get in contact with the diseased areas of the lung, as they are applied directly through the trachea. Our results are in line with those reported by Antunes et al., who also observed a greater therapeutic effect of i.t. versus i.v. administration of MSC (of bone marrow origin) in the elastase-treated mouse, reporting a further reduction of alveolar hyperinflation and collagen in i.t.-treated mice (however, change of macrophage phenotype M1 to M2 was only observed in i.v. applications of BM-MSC) [25]. We observed reduction in elastase-induced lung emphysema after MSC application; however, we have used only the quantification of emphysematous area as a tool to measure the therapeutic effect; therefore, other readouts, such an expression of epithelial and elastic fibre markers, as well as analysis of immune cell populations, would be required to further elucidate the molecular and cellular mechanisms behind the UC-MSC i.v. versus i.t. therapies.

Accumulating evidence has suggested that the therapeutic effect of MSC is mainly mediated via secreted factors [30,31]. Accordingly, Katsha et al. observed that engraftment of pre-stained MSC significantly decreased after 14 days of administration into the elastase-treated mouse; therefore, he assumed that the therapeutic effects were rather exerted by the release of paracrine factors such as hepatocyte growth factor (HGF), epithelial growth factor (EGF), and secretory leukocyte protease inhibitor (SLPI) [32]. When analysing the UC-MSC treatments, we found human mitochondrial DNA in the lung tissue from MSC-treated mice and further detected human mitochondria in immunostaining of histological samples, which suggests that MSC mitochondria might have been transferred at some point and have remained within the injured areas. In fact, one of the mechanisms proposed for the paracrine-mediated repair of MSC is the transfer of mitochondria [19]. For instance, human-induced multipotent MSC was found to transfer active mitochondria to lung cells, which protected them against oxidative stress and reduced lung injury and inflammation in vivo [33]. It has been reported that mitochondria transfer from MSC can rescue and restore functional properties in the recipient cells [19]. In fact, Ahmad et al. demonstrated that mitochondria were transferred from MSC (applied i.t.), thus increasing ATP levels and reversing rotenone-induced airway epithelial injury in mice [34]. However, despite the contribution of MSC-derived mitochondria after MSC administration to mediate repair is clear [35], the question whether the sole transplantation of human purified mitochondria from MSC would have the same beneficial effect deserves further investigation. Even though our data might suggest that MSC may have died leaving the mitochondria around, we must be cautious when making such a statement because, even conceding that the droplet digital PCR is one of the more sensitive methods for detecting biological material, we should also emphasize that we were injecting only a few thousand human cells in a full mouse lung.

Recruitment of the MSC or homing to the injured areas to mediate repair is another key mechanism of MSC biology [36]. Our protein array revealed that UC-MSC treatment in the elastase-treated mice upregulated the levels of chemokine CCL21, which is known to promote the homing of MSC to the wounds [37], along with an increase in the growth factor FGF1, which is secreted by MSCs to mediate repair [38]. Interestingly, there was an attempt to restore the levels of ADIPSIN in the elastase-treated mice upon MSC treatment, an adipokine that is found upregulated in COPD [39]. In summary, our results indicate that the therapeutic effect of UC-MSC administration in the elastase model is most likely being mediated through mitochondrial transfer, migration of MSCs, and modulation of the immune response.

Immunomodulation is one of the key mechanisms proposed for MSC-mediated repair in respiratory diseases [40,41]. In our in vivo experiments, we used an immunodeficient (nude) mouse model, which presents a clear limitation to study the immune response that takes part during pathology; therefore, we do not know to what extent the nature of the model might have influenced the mechanistic response that we observed upon UC-MSC treatment. We also used this model to test the therapeutic effect of COPD vs. non-COPD AD-MSC and, despite the abovementioned limitations, we could demonstrate that AD-MSC from COPD subjects are equally capable of restoring lung damage. We found a significant reduction in the emphysematous areas in MSC-treated mice; however, other experimental models using immunocompetent mice must be required to elucidate the MSC effect regarding the inflammatory response.

The sample size in our in vivo experiments was limited due to the difficulty of obtaining lung tissue, especially for those coming from lung biopsies, which happened only in limited scenarios. The fact that primary cells such as human AD-MSC can only be grown in suitable conditions in passages not higher than 10–12 did not allow us to use cells from the same subjects for experiments long in time. Nevertheless, even with these limited N numbers, we could perform experiments that support and lead to significant conclusions.

In conclusion, we have compared for the first time to our knowledge the in vitro properties and therapeutical efficiency of COPD vs non-COPD human MSC of adipose origin, thus reporting that, despite COPD AD-MSC presenting some in vitro functional differences, both cohorts were capable of reducing emphysema in our elastase mouse model by a >50% reduction of the emphysematous areas. These results demonstrate for the first time that COPD does not seem to cause a major systemic effect in the regenerative potential of the adipose MSC niche, therefore making suitable the autologous treatment in COPD patients. We also found therapeutic potential of UC-MSC applied in our model, either i.v. or i.t., which is most likely due to mitochondrial transfer, immunomodulation, and homing to the injured areas. Our data support the use of MSC from adipose and umbilical origin as two minimum invasive sources for the treatment of COPD.

## 4. Materials and Methods

### 4.1. Patient Data

A total of 13 non-COPD and 12 COPD subjects were included in the study. MSC of adipose origin were obtained from the University Hospital of Son Espases (Mallorca, Spain) from individuals that required surgical intervention for lung resection or from healthy donors that died from reasons other than lung disease. Individuals with advanced lung cancer (lymph node involvement and/or metastases) or any other cancer regardless of stage, liver disease, chronic kidney disease, thyroid dysfunction, severe heart disease, and asthmatics were excluded. Patients’ clinical data are displayed below in Table 1. All individuals signed informed consent. The project was approved by the Ethics Committee of the Balearic Islands (CEIC) in Spain (IB 1991/13 PI, 30 of January 2013). MSCs of umbilical cord origin were obtained from the IdISBa Biobank of Palma (Mallorca, Spain) from a total of 10 cord blood donors, healthy non-smoker mothers, after signing informed consent approved by the CEIC (IB 745/06 PI, 27 of October 2010).

### 4.2. Isolation and Characterization of Human MSC

Samples and data from patients included in this study were provided by the CIBERES Pulmonary Biobank Consortium, a network currently formed by twelve tertiary Spanish hospitals (www.ciberes.org), detailed in the Acknowledgements section, integrated in the Plataforma ISCIII Biobancos y Biomodelos. Samples were processed following standard operating procedures with the appropriate approval of the Ethics and Scientific Committees. Human MSC were isolated from fresh human subcutaneous fat tissue (AD-MSC) or fresh umbilical cord Wharton’s jelly (UC-MSC) using the following standardized protocol: tissues were first minced and digested with a collagenase (Sigma-Aldrich, St. Louis, MO, USA) for 30 min at 37 °C. PBS was then added to dilute and decrease the enzymatic activity, and the cell suspension was serially filtered through 100 µm and 75 µm cell-strainers (352360, Corning, Somerville, MA, USA). To remove cell and tissue debris and to completely inactivate the digestive enzyme, the sample was washed in PBS three times at 600 xg (Eppendorf centrifuge 5810R). The pellet containing a heterogeneous population of cells (stem-vascular fraction; SVF) was plated on cell culture-treated flasks. Cells were cultured for 4–6 days under standard conditions 37 °C, 5% CO_2_, 98% humidity in αMEM medium (L0475-500, Biowest, Nuaillé, France) supplemented with 10% FCS (S1810-500 Biowest, France) and 1% penicillin/streptomycin (PS-B, Capricorn Scientific GmbH, Ebsdorfergrund, Germany). UC-MSCs were cultured with 10% human platelet lysate or 20% FBS as supplement. After 4 days, non-adhered cells were removed by changing the media. Cells were maintained and grown until 90% of confluence by changing the medium 2–3 times weekly. After 2–3 weeks, the cells (passage 1–2) were detached by trypsin/EDTA 1× (L0930-100, Biowest, Nuaillé, France). Viability and mesenchymality were evaluated as follows: MSC were differentiated (Human Mesenchymal Stem Cell Functional Identification Kit, SC006, R&D, Toronto, ON, Canada) and the phenotype of cells were identified by flow cytometry analysis (Mesenchymal SC analysis kit, 562245, BD Bioscience, Satndford, CA, USA). MSC were positive for CD105 PerCP-Cy™5.5/CD73 APC/CD90 FITC, but negative for CD45/CD34/CD11b CD19/HLA-DR PE. Certified MSCs were used for the experiments in passages 4 to 10.

### 4.3. MSC Differentiation Assays

Differentiation assays were performed following the instructions from the Human Mesenchymal Stem Cell Functional Identification Kit (SC006kit, R&D, Ottawa, OK, Canada). MSCs were cultured for 21 days in 96 multiwell plates (165306, NUNC, Thermofisher Scientific, Waltham, MA, USA). The differentiated cells were identified by immunostaining with anti-osteocalcin (osteoblasts) and anti-FABP4 (adypocites) antibodies (both from R&D, Toronto, ON, Canada, SC006). Quantification was completed using near infrared IRDye^®^ conjugated secondary antibodies by the Odyssey system (Li-Cor Biosciences, Lincoln, NE, USA) following standard protocols for In-Cell Westerns. Briefly, the differentiated cells were fixed in 4% formaldehyde in 1X PBS for 20 min at room temperature (RT), permeabilized, and washed five times for 5 min with 1X PBS containing 0.1% Triton X-100. Primary antibodies were incubated overnight at 4 °C, and after three brief washes with PBS, an IRDye secondary antibody and a dilution of *DRAQ5*™ (BioStatus, Cincinnati, OH, USA), which binds to DNA and works a normalization agent as a measurement of cell density, were incubated for 1 h at room temperature. After three additional washing steps, the plate was read at 700 and 800 channels using the Odyssey CLx (Li-Cor Biosciences, Lincoln, NE, USA).

### 4.4. Cellular In Vitro Treatments

#### 4.4.1. Cigarette Smoke Medium (CSM)

The smoke of two cigarettes (10 mg nicotine) was gassed/“bubbled” (50 mL syringe: one puff 10 mL/30 section) into 10 mL of 37 °C pre-warmed HBSS 1× (10× conc.; L0605, Biowest, Nuaillé, France). A neutral pH 7.4 in the smoked solution was adjusted with 2% hypochloric acid. The resulting (100% concentrated) Cigarette Smoke Medium (CSM) was filtered sterile by using a 0.22 µm syringe filter unit (SLGP033RS, Merck Millipore, Burlington, MA, USA). The concentrated CSM-HBSS solution was then diluted in 1%FCS, 1% P/S αMEM medium to be used as a 10% or 5% working solution to stimulate MSCs during cell attachment experiments.

#### 4.4.2. VEG-121 Stimulations

Cells were stimulated with VEGF-121 (#100-20A, PeproTech, London, UK) diluted in 1%FCS, 1% P/S αMEM medium at the concentration of 10 ng/mL. The same medium alone was used as control and 10%FCS 1%P/S medium was used as positive control.

### 4.5. Cellular Migration and Attachment Assays

MSC were seeded in 96-well electronic microplates (E-Plate 96 view, ACEA Biosciences, San Diego, CA, USA) at 5.000 cells per well in 10% FBS 1% P/S αMEM medium and the cellular attachment was measured by xCELLigence Real-Time Cell Analyzer System (RTCA, Agilent, San Diego, CA, USA) was used according to the manufacturer’s instructions. Briefly, the E-Plate was first incubated in the device for 1 h and a measurement step was performed in wells containing only medium to obtain the background. Wells containing cells were treated with the following conditions in at least five replicates per group: 1% FCS (neg-control), and 10% FCS (pos. control) medium. Measurements were performed under 37 °C, 5% CO_2_, 98% humidity every 30 min for 90–120 h. Signals representing “attraction or cell-activation” were obtained by subtracting background values from the positive control. MSC migration analysis was performed using modified 16-well plates (CIM-16; ACEA Biosciencies, Barcelona, Spain) with MSC seeded in duplicates at 20.000 cells per well. Prior to each experiment, cells were deprived of FBS during 24 h in 1% FCS αMEM medium. Initially, 160 µL and 30 µL of media were added to the lower and upper chambers, respectively, and the CIM-16 plate was locked in the RTCA DP device inside the incubator for 60 min to obtain the background signal. To initiate the cell migration, 160 µL of control medium or migration-stimulating media (10% αMEM or VEGF-121in 1% FCS 1%P/S αMEM) was added to the medium in the lower part of the CIMplate. The Cell Index detection was performed every 5 min for 4–13 h. All data were analyzed with RTCA software (vs. 1.2.1).

### 4.6. Droplet Digital PCR

DNA was isolated from mouse lungs inoculated or not with human MSC using the DNeasy Blood & Tissue Kit (Qiagen, Madrid, Spain). The ddPCR reaction was performed using the ddPCR™ Supermix for Probes (Bio-Rad, Barcelona, Spain) and following the manufacturer’s instructions for DNA > 66 ng and digested with a restriction enzyme (Sma I). All reactions were performed in multiplex within the same tube with FAM (human) and HEX (mouse) fluorophores under an annealing temperature of 56 °C. The presence of human, genomic, and mitochondrial DNA was detected in whole mouse lung homogenates by Droplet Digital PCR (ddPCR, Bio-Rad, Hercules, CA, USA) using the following primers/probes:

Genomic Human (FAM);

Hu-RPP30-Probe: FAM-CTGACCTGAAGGCTCT-MGBNFQ,

Hu-RPP30-F: GATTTGGACCTGCGAGCG,

Hu-RPP30-R: GCGGCTGTCTCCACAAGT

Mitochondrial Human (FAM);

ddmtDNA-Probe:FAM-CGCTGTAGTATATCCAAAGACAACCATCATTCCCCC-MGBNFQ,

ddmtDNA-F: CCCTGACCCCCATGCCTCA,

ddmtDNA-R: GCGGTGTGGTCGGGTGTGTT

Genomic Mouse (HEX);

M-RPP30-Probe: HEX-CCGGGGCAAGAGTGACTGAA-MGBNFQ,

M-RPP30-F: GCTCACCACTAAGCCATAGG,

M-RPP30-R: ACAGGGAAGATGGTACTGGT

### 4.7. MSC Staining and Image Acquisition

MSCs were stained with the infrared dye CellVue™ NIR815 following the manufacturer‘s protocol (88-0874-16, Thermo Fisher Scientific, USA). MSCs were then administrated into the mouse following the routes and schemes described in the results section. Images were obtained and the end of the experiments in full animals using the *Pearl*^®^ Impulse Imager or in organs and histological sections using the Odyssey^®^ CLx and the Image Studio™ software v.5.2 (LI-COR).

### 4.8. Animal Elastase Emphysema Model

The animal procedure was evaluated by the Ethical Committee with the application code CEEA231114, and authorized by the “Conselleria de Medi Ambient, Agricultura i Pesca” of the Government of the Balearic Islands (study number 2015/01/AEXP). All animals received human care in compliance with the European directive 2010/63/EU and the Spanish Royal Decree 53/2013 on the protection of animals used for scientific purposes. As we noticed an immunoreaction towards human MSC in immunocompetent BALB/c mice, we decided to develop an elastase model using the *Swiss Nude* (*Crl*:*NU*(*Ico*)-Foxn1^nu^) strain, which has a deficient immune response, lacking T lymphocytes and does not generate cytotoxic effector cells. Elastase was administrated by intratracheal inoculation with a probe (softss-20S-25, Biogen, Madrid, Spain) and concentration/dose had to be significantly increased by up to four doses (E1250, Sigma-Aldrich, 0.7 IU/mouse; approximately 0.028 IU/g, 50 μL total/day) in order to achieve similar results to those previously published. UC-MSC or AD-MSC was applied intravenously or intratracheally twice (250,000 cells/≤75 μL saline/lung), the first time being 2 days after the last elastase inoculation, and the second, 3 days after the previous one. Animals were euthanized 4 days later by anesthetic overdose, and their lungs were harvested and processed for histology by hematoxylin/eosin labelling (H&E). H&E sections were scanned by Cell Observer microscope (Zeiss) using tiles to obtain a full area of each of the tissue slices. The emphysematous area was measured with Digimizer Image Analysis Software (MedCalc Software), normalized to its corresponding total area of the same section, and an average of the data. At least three mice were pooled per treatment.

### 4.9. Protein Array

Lungs of at least three mice per group (Control, Elastase & Elastase + MSC) were dissociated using a gentleMACS Dissociator (Miltenyi Biotec), protein was measured with the DC™ Protein Assay (Bio-Rad, Hercules, CA, USA), and equal protein concentrations per group were analyzed following the Proteome Profiler™ Array (Mouse XL Cytokine Array Kit, R&D Systems ARY028) recommended protocol (http://bit.ly/2lAbFda) assessed on 1 May 2022. The membrane spots were quantified using the Image Studio software (Li-Cor Biosciences).

### 4.10. Statistics

Data are presented as mean ± standard deviation. Student’s *t*-test was used to compare means between two experimental groups. Analysis of variance one-way ANOVA followed by Bonferroni post hoc test was used to analyze the effect of the treatment on each group individually, whereas two-way ANOVA followed by Bonferroni post hoc test was applied to assess the effect of the treatment among the experimental groups. A *p*-value of <0.05 was considered statistically significant. All statistics were carried out using GraphPad Prism (GraphPad Software v.8.0.2).

## Figures and Tables

**Figure 1 ijms-24-05813-f001:**
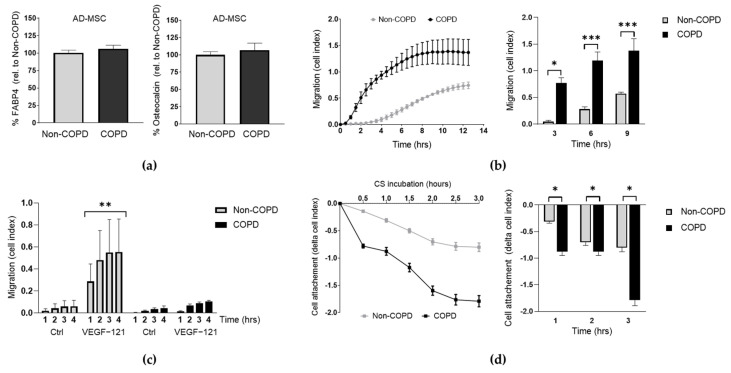
COPD derived AD-MSC have impaired migration in response to VEGF and cigarette smoke stimulus. (**a**) Analysis of the cell differentiation capability of AD-MSC from non-COPD and COPD patients into adipocytes (FABP4) and osteoblasts (osteocalcin) upon 21 days of stimulation. Data presented as % protein levels relative to non-COPD. (**b**) Migration assay of AD-MSC from non-COPD (*n* = 3) and COPD (*n* = 4) patients at baseline conditions. Two-way ANOVA, Bonferroni post-test. (**c**) Migration assay of AD-MSC from non-COPD (*n* = 4-6) and COPD (*n* = 3) patients in response to VEGF-121 at the time points indicated. Two-way ANOVA, Dunnet’s post-test. (**d**) Cell attachment of AD-MSC from donor and COPD patients in response to CSM at real time during 3 h. Pool of 3 patients per condition. Student’s paired *t*-test. *p* values: * *p* < 0.05, ** *p* < 0.005, *** *p* < 0.0005. Abbreviations: AD-MSC (adipose tissue MSC), control (Ctrl), CS (cigarette smoke).

**Figure 2 ijms-24-05813-f002:**
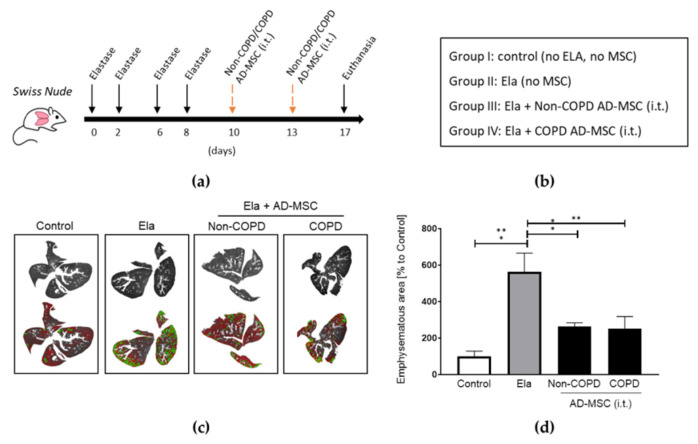
Human AD-MSC derived from COPD patients were equally efficient as those from non-COPD in reducing elastase-induced lung emphysema. (**a**) Timeline in days of the elastase and MSC treatments, and (**b**) groups of treatments included. (**c**) Digital evaluation of emphysematous areas of the lungs and (**d**) quantification and statistics. Treatments were performed intratracheally (i.t.). *n* = 3 per group. 1 way ANOVA, Tukey post-test, * *p* < 0.05, ** *p* < 0.005.Abbreviations: AD-MSC (adipose tissue MSC), Ela (elastase), i.t. (intratracheally).

**Figure 3 ijms-24-05813-f003:**
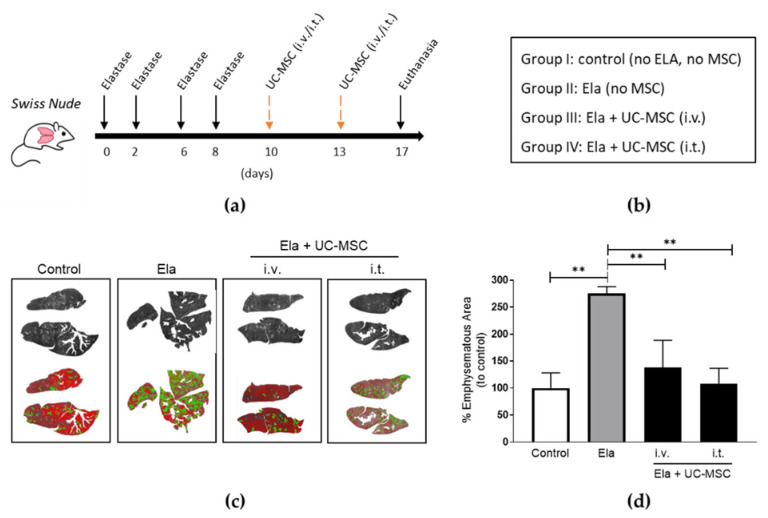
Human UC-MSC administrated either intravenously (i.v.) or intratracheally (i.t.) reduced lung emphysema in elastase-treated mice. (**a**) Schematic timeline and (**b**) groups of the elastase and MSC treatments included. Images of digital evaluation (**c**) and quantification and statistics (**d**) of the emphysematous areas of the murine lungs. One-way ANOVA, Tukey post-test ** *p* < 0.005. Abbreviations: AD-MSC (adipose tissue MSC), UC-MSC (umbilical cord MSC), Ela (elastase), i.v. (intravenously), i.t. (intratracheally).

**Figure 4 ijms-24-05813-f004:**
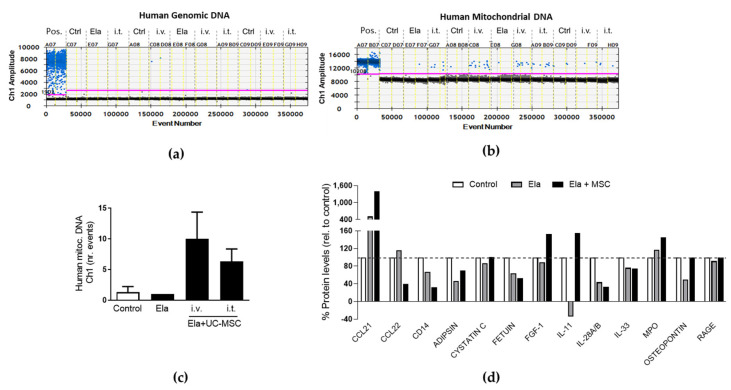
Human mitochondria presence and proteomic cytokine and chemokine profile upon UC-MSC treatment in the elastase model. Analysis of mitochondrial (**a**) and genomic (**b**) human DNA by droplet digital PCR in lungs from control (Ctrl) and elastase-treated (Ela) mice treated with human UC-MSC applied intravenously (i.v.) or intratracheally (i.t.), and (**c**) quantification of the mitochondrial human DNA. (**d**) Protein array of lung homogenates from mice subjected to the same treatments. Pool of at least 3 mice per condition. Abbreviations: UC-MSC (umbilical cord MSC), Ela (elastase), Pos. (positive control), Ctrl (control), i.t. (intratracheally), i.v. (intravenously).

**Table 1 ijms-24-05813-t001:** Patients’ clinical data.

Variable	COPD	Non-COPD	Sig. (*p*)
N	12	13	
Age	64.75 ± 6.01	58 ± 12.4	0.101
Gender (%) m/f	10 (83.33)/2 (16.66)	8 (61.53)/5 (38.46)	0.292
**Smoking history**			
smoker %	6 (50)	7 (53.84)	
exsmoker %	6 (50)	4 (12.5)	0.469
nonsmoker %	-	2 (6.25)	
smoking index (*p*/y)	92.75 (40–400)	32.13 (3–65)	0.152
**Spirometry**			
FVC (L)	2.95 ± 0.68	3.95 ± 0.71	**0.0015 ***
FVC (% predicted)	80.7 ± 14.25	99.33 ± 2.87	**0.0001 ***
FEV1 (L)	1.48 ± 0.36	2.89 ± 0.61	**<0.0001 ***
FEV1 (% predicted)	55.55 ± 12.49	91.67 ± 8.99	**<0.0001 ***
FEV1/FVC %	50.43 ± 10.59	69.94 ± 7.91	**<0.0001 ***
**GOLD Stage**			
GOLD I	1 (8.33)	-	
GOLD II	8 (66.66)	-	
GOLD III	3 (25.0)	-	
GOLD IV	0	-	

Abbreviations: p/y, packs/year index; FVC, forced vital capacity; FEV1, forced expiratory volume in 1 s; GOLD, Global Organization for Pulmonary Disease; * Significant *p* < 0.05, Student’s *t*-test.

## Data Availability

Not applicable.

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
