# Peer review of "Mesenchymal Stem Cells from COPD Patients Are Capable of Restoring Elastase-Induced Emphysema in a Murine Experimental Model"

_ijms, 2023, doi:10.3390/ijms24065813_

Round 1
Reviewer 1 Report
Very interesting work with a very good quality of pesentation. I would suggest the publication of the study. I have no comments to the authors.
Author Response
The authors are very grateful to the reviewer for accepting reviewing our work and for finding our results interesting and valuable for the scientific community. We are also thankful to the reviewer for such a positive comment.

Reviewer 2 Report
In this manuscript, Rio and Jahn et al. test therapeutic efficiency of adipose tissue- mesenchymal stem cells (AD-MSCs) and umbilical cord- MSCs (UC-MSCs) in alleviating emphysema in an immunodeficient mouse model of elastase induced emphysema. The manuscript is well written, easy to read, experiments are well explained and results clearly outlined and described. Overall this manuscript is a great addition to the community's knowledge for use of MSCs in therapy of lung diseases. There are a few concerns stated below that if addressed will greatly strengthen the work:
1. The authors repeatedly mention pluripotency as a characteristic of MSCs but that is not entirely definitive. Only a subset of MSCs express OCT-4 and hence are pluripotent. It might benefit the authors to replace pluripotency with multipotency.
2. The authors use AD-MSCs using i.t. delivery but then use UC-MSCs using i.t. or i.v. delivery. Hence it is difficult to adjudge how AD-MSCs and UC-MSCs perform when compared to each other. This is relevant because in lines 186-188 the authors compare these 2 separate experiments which I agree is good information to have. It would be helpful for the readers to know how COPD- AD-MSCs, non-COPD AD-MSCs and UC-MSCs perform compared to each other in the migrational and functional tests run in Fig 1.
3. It is intriguing that the lungs have human mitochondrial DNA but no genomic DNA. While this suggests that the MSCs may have died leaving mitochondria around, can the authors determine if these human mitochondria are within mouse lung cells or are they present extracellularly in the airspace or interstitium? Can delivery of purified human mitochondria from MSCs (or any human cell really) treat the murine emphysema?
Author Response
The authors are very grateful to the reviewer for accepting reviewing our work and for finding our results interesting and valuable for the scientific community. We think that his/her comments will significantly improve the manuscript, thus we have revised and adapted the manuscript accordingly. Please find the answers to every point in the word document attached.

Reviewer 3 Report
This is a timely study on the functionality and effect of mesenchimal stem cells on emphysema. The authors demonstrate results on in vitro and also in vivo experiments. and show that mesenchimal cells from COPD and non-COPD subjects have similar efficacy in reducing experimental emphysema and they suggest that these findings provide background for potential autologous use of mesenchimal cells in COPD.
Comments:
The manuscript is well-written, easy to follow well documented. Positioning current treatments of COPD in the introduction please have a say about mortality (according to the recent GOLD triple therapy may even decrease mortality). Obviously, they do not change lung destruction- so the reasoning of the authors is valid to do research on this part.
Sample size in some cases is very low (n= 3, or 4-6 in some measurements). Appreciating the limits of improving numbers and the fact that despite the low numbers there are significant differences between groups, limitations due to low sample sizes needs discussion.
Please, correct decimal points in line 115 (changes commas to dots) in the Table around lines 373-379).
In the Table % of smoker in non-COPD is given as 21,88 with number 7 out of total number of non-COPD patients of 13. That cannot be right. The authors need to check all these numbers in the Table, and correct as necessary.
Please discuss the potential effect of difference in smoking history between COPD and non-COPD patients.
Author Response
The authors are very grateful to the reviewer for accepting reviewing our work and for rising such important questions. We think that his/her comments will significantly improve the manuscript, thus we have revised and adapted the manuscript accordingly. Please find the answers to every point in the word document attached.

Round 2
Reviewer 2 Report
All my comments have been addressed.